# OpenReview forum: "More Capable, Less Cooperative? When LLMs Fail at Zero-Cost Collaboration"
_ICML.cc/2026/Conference — ICML 2026 regular_

### Official Review · Reviewer_o2c5 · 2026-02-24

**Soundness:** 3
**Presentation:** 3
**Significance:** 4
**Originality:** 4
**Overall Recommendation:** 5
**Confidence:** 4

**Summary:**

Following many studies in LLM cooperation, the paper presents a cooperative environment where cooperation has no cost but provides a direct benefit to others and the LLM society as a whole, and studies if LLMs cooperate in this trivial environment. The paper finds that many LLMs still fail to cooperate, despite direct instructions to do so. Further studies are then conducted to understand how to promote cooperation in these settings, and what is the reasoning behind agents’ decisions.

**Compliance With Llm Reviewing Policy:**

Affirmed.

**Final Justification:**

I have updated my original score from a Weak Accept (4) to an Accept (5) following the rebuttal. The authors originally presented a novel idea with interesting results, however, the results were not extensively validated. Having raised questions regarding this issue, the authors did new experiments directly addressing my concerns, and as such I am confident the camera-ready will be a strong submission.

**Key Questions For Authors:**

Q1: One of the main ideas of the paper is that many cooperation problems have no cost to the helper. However, this is not entirely clear by some of the examples listed (for example, in lines 100-101 right column). For example, open science, from the perspective of a publisher naturally has different payoffs, and from the perspective of the author often incurs higher publishing costs and, depending on the field, often lower prestige. Similarly, public documentation is often very labor intensive. Could you elaborate in what way are these cooperation problems with negligible costs, how often these scenarios exist, and from whose perspective are the costs negligible as an actor? There seems to be a general reasoning that negligible costs imply no cost at all or a much lower cost in comparison to the benefit, but this requires a formal value analysis and omits discussions about absolute values (e.g. spending 10 euros to give someone 100 euros of value or spending 1000 euros to give someone 10000 euros of value offer the same proportion but at different absolute costs).

Q2: Given that agents maintain a history of the past actions, can it be that agents punish each other for not sharing information, leading to low cooperation? There is a large corpus studying costly punishments which can be used. In this case, it could also be interesting to study an inverted scenario where agents may only punish each other for no benefit or cost to see if similar anti-social behaviors happen.

Q3: The result showing that auto-request decreased the performance of o3, o3-mini and GPT 5-mini was quite surprising and I think requires a deeper explanation since the task with auto request is quite trivialised and perhaps even more reflective of the main message of the paper than the baseline experiment. Could you provide any insights or clarifications about why this drop in performance occurred, and provide more results regarding if this was delayed sending due to other limitations of the models or a real drop in cooperative intentions?

**Limitations:**

The paper should better explain the limitations of their methodology, particularly regarding the environment’s theme and complexity and the number of LLMs used (given that it omits some very popular models such as Llama and Gwen).

**Strengths And Weaknesses:**

General appreciation

The paper presents surprising results that recontextualize previous works on LLM cooperation. It suggests an internal lack of cooperative incentives in LLMs rather than an environment or coordination problem as it is typically studied. These findings are well supported with reasoning analysis, a decent number of LLMs tested, a set of interventions to improve cooperation, and more. The environment appears mathematically sound. Nevertheless, more work should be put into justifying the necessity of such a complex and thematic environment, and in making clear how the results relate to previous works in LLM cooperation.

Major issues

The paper encodes a substantially complex environment. While the limitations of agents regarding implementation is studied, it would still be very relevant to see if this problem is present in less complex environments. For example, if agents were instead playing classic game theoretical games such as the donation game (one sided prisoner’s dilemma, but set to a cost of cooperation to 0) or a public goods game. This would allow for an evaluation under a minimal environment and a much easier connection with prior works. The idea of the paper is very relevant and original, yet the way it was implemented makes these connections with previous works hard.

Similarly to the point above, it is unclear how much the framing of the environment, based on an enterprise setting, is providing agents with a context that suggests competition instead of cooperation. As such, the main conclusions of the paper may not hold if the environment was instead, for example, based on a shared community farm or a setting where agents could allocate resources such as food.

While this work shows that many models indeed do not cooperate even when it is the rational option, the work should better discuss how this integrates with findings from other works. Many previous works show that LLMs do cooperate in some cases, even when there is a personal cost to do so, however, this directly contradicts the findings of other papers and this discussion is not present both in the discussion section and in the related work. It should be made clear if this a difference in environment, or because the payoff structure is different. Without a broader integration of the findings, the paper seems to apply only to this environment.

Minor issues

Lines 20-23 right column, from the abstract, are phrased quite confusingly and somewhat ungrammatically. It also appears to create a distinction between cooperating and helping which is not made clear.

In Figure 1, the squares with crosses below Agent 1 and Agent 3 should be more explicit. One would imagine they are the missing market data, but there are only 4 boxes while agent 1 requests region 5.

In Line 322 right column, a limitation of the reasoning analysis is stated, but this should ideally be accompanied by another metric or reason to understand then why GPT-4.1-mini would perform poorly.

In Figure 5, the y-axis ticks and x-axis model names should be made larger for easier readability. A similar issue is present in Figure 4, with the origin of the axis (bottom left) having overlapping ticks.

Other issues with the paper are expressed in the questions section, as they require clarifications.

---

> ### Author Rebuttal · Authors · 2026-03-31
>
> We thank the reviewer for their thoughtful engagement with our work.
>
> ## Additional Environment Validation
>
> This is an important concern. We chose the original multi-agent setting deliberately because it lets us study sender-neutral helping in a setting that more closely resembles real-world agent deployments. Our decomposition results already suggest the baseline failures are not due only to confusion about the mechanics: under Auto-Fulfill, cooperation-limited models (o3, o3-mini, GPT-5-mini) exceed 90% of optimal, indicating that they can navigate the environment well when sharing bottlenecks are removed.
>
> Nevertheless, to directly address generalizability, we evaluate select models in a second, simpler environment where agents grant temporary resource access to unblock time-sensitive deliverables; unmet requests expire, so delays destroy value permanently. **The same qualitative pattern persists**:
>
> | Model | Tasks Completed | Messages/Task | Gini Coefficient |
> | :--- | :--- | :--- | :--- |
> | **o3-mini** | 36 (39.1%) | 8.17 | 0.1611 |
> | **o3** | 21 (22.8%) | 16.92 | 0.0652 |
> | **Claude Sonnet 4** | 69 (75.0%) | 3.91 | 0.3385 |
> | **Perfect Baseline** | 92 (100.0%) | 3.15 | 0.0174 |
>
> Relative to the perfect baseline, Claude 4 Sonnet achieves 75.0% of optimal performance, o3-mini 39.1%, and o3 22.8%; o3 also remains substantially less communication-efficient. This suggests the main finding is not an artifact of one specific testbed, but reproduces in a zero-cost cooperation setting with a different mechanism and temporal structure.
>
> ## Prior Work
> We appreciate the reviewer's concern that prior work shows LLMs cooperating even at personal cost. The key distinction is the incentive regime: those settings involve reputation effects, private costs, or coordination risk that give models instrumental reasons to cooperate. Our zero-cost design intentionally removes such confounds to establish a lower bound. Recent concurrent work actually supports our findings, and a detailed reconciliation with the literature is in our response to **Reviewer Z3rN**.
>
> ## Enterprise framing affecting results
>
> Thank you for raising this point. To verify that our findings are not caused by superficial artifacts, we conducted the following prompt ablations:
>
>
> (A) Removing corporate framing,
>
> (B) Replacing all economic language with neutral/prosocial terms, and
>
> (C) Rephrasing the cooperative instruction itself.
>
>
> Our findings persist, and full results can be found in our response to **Reviewer TDtb**.
>
> ## Q1. Zero-Cost Framing
>
> The reviewer raises a fair point: for humans, open science and public documentation involve real effort and costs. We agree, and will revise the framing. The real-world examples motivate the multi-agent setting, not the cost structure. The zero-cost design captures the cost structure of an LLM agent deployed in such roles, where **sharing is a copy operation**: the sender retains the piece, incurs no computational or strategic cost, and R_i is unaffected, which helps establish a lower bound on cooperation. We will sharpen this distinction between motivating setting and formal cost structure in the revision.
>
> ## Q2. Can agents punish each other?
>
> This is a very interesting question. Our evidence suggests punishment dynamics are not the primary driver. In the Auto-Request condition, requests are automated each round optimally, yet cooperation-limited models (o3, o3-mini, GPT-5-mini) still complete <20% of optimal tasks, and **withholding persists even absent any provocation**.
> Furthermore, agent reasoning analysis reveals proactive strategic language ("leverage," "bargaining chip") rather than retaliatory framing, and **performance actually improves with longer horizons** (Table 9), contrary to punishment-spiral predictions. Nonetheless, studying costly punishment in LLM multi-agent settings is a promising direction we will discuss.
>
> ## Q3. Decrease in performance for Auto-Request
>
> We agree with the reviewer that this condition is especially diagnostic. By automating requests, it trivializes the demand side, so any remaining shortfall must come from the sending and submission. Strong models (Gemini-2.5-Pro, Sonnet 4) performing near-optimally here confirms the condition works; the drops for o3, o3-mini, and GPT-5-mini therefore **directly evidence a failure to share even when trivial**. These models spend 3–5× more actions on redundant messaging than on productive information-sending, compared to <1× for the strong models.
>
> Importantly, the failures are distinguishable and answer the reviewer's question about delayed sending vs. non-cooperation. **o3-mini** shows the largest task drop but near-zero withholding reasoning (0.1%), dominated by inefficient execution, a protocol-adaptation failure. **o3** mixes misadaptation with leverage-preserving reasoning. **GPT-5-mini** shows the strongest conditional non-cooperation, with explicit trade-first reasoning suppressing timely sharing.

---

> > ### Author Rebuttal · Reviewer_o2c5 · 2026-04-02
> >
> > I appreciate the author's replies, which have largely addressed my concerns. The presence of a second framing makes the paper's findings much stronger, although I still think it would benefit from a purely game-theoretical setting such as a very simply defined donation game (without any real world framing).
> >
> > Finally, I would like a better clarification regarding your answer to my first question:
> >
> > First, are the results independent of the scale of the benefit? For example, would the results be different if instead every benefit would be multiplied by 100 (increasing the social benefit), or divided by 100 (potentially decreasing competitiveness).
> >
> > Second, given that LLMs (perhaps) know they are not capable of executing these decisions, could they be acting in the capacity of advisors and assume that a human user will actually have to take the cost of cooperating? That is, when prompting the LLM to make a decision, can they be expecting that a human will ultimately be executing this decision and as such are considering a possible cost (even if just time cost or token cost) to the human?

---

> > > ### Author Response · Authors · 2026-04-07
> > >
> > > We thank the reviewer for the thoughtful follow-up. We are glad the second environment helped strengthen the broader finding. The two follow-up questions sharpen an important point in our earlier response about whether models are actually representing the setup as sender-neutral helping. We address each directly below.
> > >
> > > ## Q1. Sensitivity to the scale of the benefit
> > >
> > >
> > > This is a very useful clarification. Under the formal payoff structure, uniformly multiplying all task benefits by a positive constant **does not change which policy is optimal**: helping remains costless to the sender and continues to improve collective welfare. Thus, if behavior changes under rescaling, that would reflect model sensitivity to numerical salience or task representation rather than a change in the underlying incentive structure itself. We report total tasks completed under four rescalings of the task benefit:
> > >
> > > | Model | **$100** | **$1,000** | **$10,000 (default)** | **$100,000** |
> > > | :--- | ---: | ---: | ---: | ---: |
> > > | o3-mini | 100 | 118 | 103 | 114 |
> > > | o3 | 32 | 17 | 34 | 38 |
> > > | Claude Sonnet 4 | 135 | 141 | 132 | 127 |
> > >
> > > Empirically, **the qualitative pattern is stable across these four scales**. Rescaling changes performance levels somewhat, but it does not produce a monotonic relationship with reward magnitude or eliminate the central cross-model gap. o3 remains consistently low across all settings (17–38 tasks), Claude Sonnet 4 remains consistently strong (127–141), and o3-mini remains substantially above o3 at every scale (100–118 vs. 17–38). This suggests that the main phenomenon is **not driven by the absolute magnitude of the stated rewards**, even though numerical framing can modulate performance at the margin.
> > >
> > > We will incorporate this clarification in the revision, since it sharpens the distinction between the formal incentive structure of the environment and the way models may internally represent that structure.
> > >
> > > ## Q2. Models as Advisors for Human Executors
> > >
> > >
> > > This is a useful alternative interpretation to examine. However, our transcript evidence does not support this explanation as the primary driver of the observed withholding.
> > >
> > > In our setup, the models are prompted and behave as direct in-environment actors, not as advisors to an external human. The agent prompt places the model in the role of an acting agent and defines executable actions; the model’s JSON output directly instantiates the simulated action. Consistent with this framing, across the full baseline transcript corpus, we do not find explicit reasoning about a human user, manual execution, API cost, latency, or token cost. In particular, across 6,816 private thoughts and 16,647 authored messages, explicit human/user/manual/API/latency/token-cost framing appears **0 times**, and we likewise find **0 instances of explicit advisor-style language**.
> > >
> > > What we observe instead is first-person in-environment action planning: **87.8%** of private thoughts contain direct-action language such as “I will send,” “I will submit now,” or “I will hold this for leverage.” This pattern holds across both high-performing and low-performing models. In other words, **the relevant difference is not who the model believes is executing the action, but which policy the acting agent adopts within the task**. Cooperative models reason about sending and submitting now; withholding models reason about leverage, reciprocity, and conditional exchange.
> > >
> > > Relatedly, when cost-like language does appear, it refers to in-world strategic considerations such as leverage, trust, or one-sided trade risk, rather than to any external human burden. We will clarify this distinction explicitly in the revision. The observed withholding is therefore better explained by **within-task strategic reasoning** than by concern about an imagined human executor’s time or token cost.
> > >
> > > Finally, we also agree that a minimal game-theoretic setting, such as a donation game, would be a valuable bridge to the classical literature, and we will note this more explicitly as a natural future extension.

---

### Official Review · Reviewer_RLXs · 2026-03-10

**Soundness:** 3
**Presentation:** 4
**Significance:** 4
**Originality:** 4
**Overall Recommendation:** 5
**Confidence:** 4

**Summary:**

This paper examines whether LLMs cooperate to generate collective benefits at negligible personal cost by isolating cooperative behavior from strategic behavior. Results show that LLMs broadly fail to meet cooperative objectives: some models actively sabotaging others. Prompting interventions (called policy-level instructions, small incentives, and limited visibility) successfully improve cooperation. Notably, the authors find that more capable models are not necessarily more cooperative, which suggests that cooperation does not emerge naturally from increased competence.

**Compliance With Llm Reviewing Policy:**

Affirmed.

**Key Questions For Authors:**

1. As noted, the decomposition experiment isolates failure modes through ablation rather than through a formal causal identification strategy. Can the authors clarify what causal claims are and are not being made, and whether the framing can be tightened?

2. It would be interesting to understand the impact of reasoning on cooperative behavior. Do higher reasoning levels (i.e. higher reasoning effort when the hyperparameter is available) lead to lower or higher levels of cooperation?

3. Will the authors release an open-source version of their evaluation framework? It would be useful for the community to be able to track the evolution of cooperative behavior as new models are released.

**Limitations:**

Yes

**Strengths And Weaknesses:**

**Strengths**
- **Environment design.** The environment is well-motivated and isolates cooperative behavior from strategic behavior. The method removes strategic complexity by making communication costless and information passing occurs at no cost or harm to any of the agents. In doing so, the authors establish a lower bound on cooperation failures.
- **Conceptual contribution.** The instruction-utility gap is a clean and important conceptual contribution.
- **Reasoning analysis.** The authors do a thorough analysis of the internal reasoning of agents (rather than just an isolated case study), which supports their clam that cooperation failures are deliberate rather than accidental. This analysis relies on the assumption that reasoning is faithful to decision-making, but the authors acknowledge this limitation.
- **Interventions.** The three interventions are well-targeted: each interventions maps cleanly onto a specific failure mode.
- **Presentation.** The paper is clearly written and well-structured with intuitive figures that effectively communicate the core findings.

**Weaknesses**
- **Homogenous agents.** All agents in a given run use the same underlying LLM. Real multi-agent deployments are likely to involve heterogeneous model populations, which could produce very different cooperative dynamics. This limitation of the results could be noted as an avenue for future work.
- **Causal analysis.** The use of the term "causal" feels somewhat overstated. While the decomposition experiment is well-designed, automating one side of agent communication does not constitute a fully causal analysis in the formal sense -- it is closer to a controlled ablation. The authors should either clarify the precise sense in which causality is claimed, or adjust the framing accordingly.

---

> ### Author Rebuttal · Authors · 2026-03-31
>
> We thank the reviewer for their positive review and appreciation of our work.
>
> ## Causal framing
>
> We appreciate this methodological push. The reviewer is right that our design does not constitute causal identification in the formal econometric sense (e.g., instrumental variables, natural experiments). We will revise the terminology, replacing "causal decomposition" with "**controlled decomposition**".
>
> We also want to clarify that the design does more than a standard ablation. Rather than simply removing a component, it intervenes on one communication channel by **setting it to optimal while leaving the other under model control**. Auto-Request holds demand constant at the perfect-play level, isolating the sharing decision; Auto-Fulfill holds supply constant, isolating requesting and submission.
>
> This means performance shortfalls in each condition are informative about a specific failure mode under controlled conditions, rather than merely observing that removing something degrades performance. We believe this interventionist logic supports controlled attribution of failure modes, and will frame it precisely as such in the revision. We thank the reviewer for pushing us to articulate this distinction clearly.
>
> ## Reasoning effort and cooperation
>
> This is a question we find genuinely exciting. Our cross-model comparisons offer suggestive indirect evidence: o3 and o3-mini share the same model family but differ in reasoning capacity, and o3 shows dramatically worse cooperation (17% vs. 50% of optimal) despite superior competence (94.9% vs. 92.1% under Auto-Fulfill). This pattern is suggestive of deeper reasoning enabling more elaborate strategic framing rather than better cooperation: o3's reasoning traces contain **39.3% hard defection language versus 0.1% for o3-mini**, with 373 instances of leverage/bargaining terminology compared to 1.
>
> A controlled reasoning-effort sweep within a single model (varying the reasoning budget hyperparameter where available) would cleanly isolate this effect and is a concrete direction we plan to pursue.
>
> ## Homogeneous agent populations
>
> We agree this is a natural next step. Heterogeneous populations would test whether cooperation failures compound or cancel across model types: for instance, whether pairing a cooperation-limited model (o3) with a competence-limited model (Gemini-2.5-Flash) produces better or worse system outcomes than either homogeneous group. Our per-model failure mode taxonomy (§4) provides the foundation for designing such experiments systematically. We will add this as an explicit future direction.
>
> ## Open-source release
>
> We plan to release the full evaluation framework, including the environment implementation, agent scaffolding, decomposition conditions, intervention configurations, and analysis scripts. We share the reviewer's view that tracking the evolution of cooperative behavior as new models are released would be valuable for the community, and will design the release to support this.

---

> > ### Author Rebuttal · Reviewer_RLXs · 2026-04-02
> >
> > Thank you for your response; I retain my positive assessment of the work and would encourage the authors to include the results of a reasoning-effort sweep in their revised paper.

---

> > > ### Author Response · Authors · 2026-04-07
> > >
> > > Thank you for the thoughtful follow-up and for confirming your assessment. We will include the reasoning-effort sweep in the revised paper; we agree that it is one of the most natural and exciting next experiments to pursue.

---

### Official Review · Reviewer_Z3rN · 2026-03-13

**Soundness:** 2
**Presentation:** 2
**Significance:** 2
**Originality:** 1
**Overall Recommendation:** 2
**Confidence:** 5

**Summary:**

This paper investigates whether LLM agents cooperate when doing so has zero cost to the individual agent and provides collective benefits. Experiments are conducted based on a turn-based, multi-agent environment where information is non-rivalrous and communication is free. Key findings reveal that more advanced reasoning models often exhibit lower cooperative performance compared to smaller models.

**Compliance With Llm Reviewing Policy:**

Affirmed.

**Final Justification:**

This paper investigates zero-cost multi-agent cooperation. After carefully reviewing the rebuttal, I am maintaining my score due to incremental novelty, statistical contradictions, and prompt-induced artifacts.

First, the authors attempted to establish a novel "capability-cooperation inversion", however, their rebuttal concedes that capability and cooperation are actually statistically uncorrelated (p=0.71), directly contradicting their central narrative. Second, the core observation that LLMs struggle to cooperate without structured incentives is a well-documented phenomenon across fully cooperative (no cost to cooperation), cooperation-neutral and mixed-motive literature, making this an incremental confirmation rather than a groundbreaking discovery.

Most critically, the observed withholding behavior appears to be an artifact of the prompt's corporate semantic framing rather than a fundamental limitation of multi-agent interactions. The authors' own ablation study reveals that simply removing economic terms (e.g., swapping "dollars" for "points") increased the o3 model's cooperation rate by ~51%. The work lacks the theoretical grounding necessary to prove these failures stem from foundational multi-agent dynamics rather than simple prompt misunderstanding (e.g., authors have shown that LLMs hallucinated a zero-sum competition which is probably based on pre-training associations with financial vocabulary).

**Key Questions For Authors:**

1. How does the proposed work fundamentally differ from established benchmarks in LLM coordination, such as [1][2][7]. Could the authors clarify the novelty and structural differences that lead to these different findings? For example, in these papers, it seems stronger reasoning models sometimes behave better on cooperative tasks.

2. Could the authors provide a systematic comparison of reasoning traces (e.g. chain of thought and private thoughts of agents) between high-reasoning models and smaller models? Explicitly identifying the reason why an agent transitions from instruction following to strategic withholding would provide necessary evidence for the paper's core conclusion.

3. Prompting strategies and agentic architectures are known to significantly impact LLM behavior [3]. Have the authors explored whether advanced prompting frameworks, such as Reflexion [6] or iterative self-correction, can mitigate the cooperative failure?

4. There is a tension between your findings and literature suggesting reasoning models cooperate more for long-term benefit [4] or strategically refuse cooperation to avoid immediate costs [5]. In a zero-cost environment, what is the formal incentive driving models to withhold? Is this a rationalized strategic choice or an objective-alignment failure?



[1] Hayler, A., Chirra, S. R., Lupu, A., Forkel, J., Sarkar, B., Feng, S., & Foerster, J. N. Zero-Shot Coordination among LLM Agents. In Workshop on Multi-Agent Learning and Its Opportunities in the Era of Generative AI.

[2] Agashe, S., Fan, Y., Reyna, A., & Wang, X. E. (2025, April). Llm-coordination: evaluating and analyzing multi-agent coordination abilities in large language models. In Findings of the Association for Computational Linguistics: NAACL 2025 (pp. 8038-8057).

[3] Dang, Y., Qian, C., Luo, X., Fan, J., Xie, Z., Shi, R., ... & Sun, M. (2025). Multi-agent collaboration via evolving orchestration. arXiv preprint arXiv:2505.19591.

[4] Zhu, S., Lin, Y., Kaistha, S., Li, W., Wang, B., Zha, H., ... & Poupart, P. (2026). Talk, Judge, Cooperate: Gossip-Driven Indirect Reciprocity in Self-Interested LLM Agents. arXiv preprint arXiv:2602.07777.

[5] Piedrahita, D. G., Yang, Y., Sachan, M., Ramponi, G., Schölkopf, B., & Jin, Z. (2025). Corrupted by reasoning: Reasoning language models become free-riders in public goods games. arXiv preprint arXiv:2506.23276.

[6] Shinn, N., Cassano, F., Gopinath, A., Narasimhan, K., & Yao, S. (2023). Reflexion: Language agents with verbal reinforcement learning. Advances in neural information processing systems, 36, 8634-8652.

[7] Lei, Y., Xie, H., Zhao, J., Liu, S., & Song, X. (2025). MSCoRe: A Benchmark for Multi-Stage Collaborative Reasoning in LLM Agents. arXiv preprint arXiv:2509.17628.

**Limitations:**

No, the authors have not adequately discussed the limitations or societal impacts.

1. The authors should discuss the limitation that zero-cost environment is a highly idealized abstraction. In practice, even without direct costs to their utility, agents may still have opportunity costs, diverting time and computational resources from their own reward-generating tasks to assist others. Neglecting these trade-offs could lead to a negative societal impact where developers deploy cooperative agents that fail or cause systemic inefficiencies in complex, real-world production environments.

2. Other limitations such as the narrow experimental domain have been detailed in the weaknesses above.

**Strengths And Weaknesses:**

Strengths:

- Multi-agent LLM coordination is an important problem.
- Good and clear writing.

Weaknesses:

1. Limited experimental testbeds and task diversity. The paper's main claim is based on a single, narrow experimental setup. To be truly convincing, the authors should validate their findings across established benchmarks like [1][2][3]. Without testing across diverse domains, it remains unclear whether these failures are fundamental to LLM reasoning or merely an artifact of the specific controlled task designed for this study.

2. Absence of ablation study on prompting strategies. The study does not sufficiently explore how variations in prompting strategies, reasoning patterns as well as agent architecture influence the cooperative outcome. Without these ablations, it is difficult to determine if the capability-cooperation inversion is an emergent property of the model's reasoning or simply an artifact of the specific prompt engineering used in the testbed.

3. It is unclear why stronger models refuse to help without a utility-based explanation for this behavior. While the authors categorize this as an instruction-utility gap, they do not reconcile why stronger models prioritize perceived strategic benefits in a technically zero-cost environment. Without a formal model of the utility gap between cooperation v.s. non-cooperation, the inversion remains a qualitative observation rather than a proven strategic phenomenon.

4. The paper's core conclusion that more capable models are less cooperative conflicts with findings in [1][2][3] and [4], which suggest that advanced models might utilize reasoning to achieve better coordination. The authors do not adequately address these discrepancies or provide a comparative baseline that explains why their specific environment triggers non-cooperative behavior while others find improved cooperation in cooperative domain [1][2][3] and even in mixed-motive domain [4].


[1] Hayler, A., Chirra, S. R., Lupu, A., Forkel, J., Sarkar, B., Feng, S., & Foerster, J. N. Zero-Shot Coordination among LLM Agents. In Workshop on Multi-Agent Learning and Its Opportunities in the Era of Generative AI.

[2] Agashe, S., Fan, Y., Reyna, A., & Wang, X. E. (2025, April). Llm-coordination: evaluating and analyzing multi-agent coordination abilities in large language models. In Findings of the Association for Computational Linguistics: NAACL 2025 (pp. 8038-8057).

[3] Lei, Y., Xie, H., Zhao, J., Liu, S., & Song, X. (2025). MSCoRe: A Benchmark for Multi-Stage Collaborative Reasoning in LLM Agents. arXiv preprint arXiv:2509.17628.

[4] Zhu, S., Lin, Y., Kaistha, S., Li, W., Wang, B., Zha, H., ... & Poupart, P. (2026). Talk, Judge, Cooperate: Gossip-Driven Indirect Reciprocity in Self-Interested LLM Agents. arXiv preprint arXiv:2602.07777.

---

> ### Author Rebuttal · Authors · 2026-03-31
>
> We thank the reviewer for their careful review.
>
> ## Prior Work
>
> We appreciate the thorough engagement with related work. **The cited works are largely consistent with our findings** once the incentive regimes are distinguished. Our finding is that capability is uncorrelated with cooperation (r=0.16, p=0.71); the contribution is this absence of reliable positive correlation, combined with a diagnostic framework explaining why models fail differently.
>
> The key distinction is the incentive regime. Prior work studies settings with private costs [5], reputation effects [4], convention formation [1], or scaffold-dependent performance [2,3,7]. Our environment removes all such confounds: **helping is costless and sender-payoff-neutral, yet agents still withhold**. This establishes a lower bound on cooperation failure not attributable to strategic rationality or coordination difficulty.
>
> [5] (which we cite) shows reasoning LLMs free-ride more in public goods games, directly aligned with our results. [4] demonstrates that reasoning models defect in finite-horizon games and cooperate only when gossip-based reciprocity makes helping incentive-compatible, paralleling our finding that a 10% bonus nearly triples o3's performance. [1] finds LLM agents consistently fail at zero-shot coordination even in simple settings. [2] shows task-dependent results where capability helps on some coordination subskills but leaves major gaps on others. **Testing sender-neutral helping, strategic withholding, or automated decompositions is unique to our paper**. We will situate our work relative to [1]-[4] explicitly in revision.
>
> ## Utility-based explanation for withholding
>
> The reviewer raises a central point: our claim is precisely that **no utility function consistent with the environment's payoff structure rationalizes withholding**. Under self-payoff R_i, sharing is payoff-neutral; under the instructed objective W=ΣR_j, sharing strictly dominates. **Withholding is therefore weakly dominated under every objective** the agent could plausibly hold, which is what makes the finding significant.
>
> To the question of whether this is a rationalized strategic choice or an objective-alignment failure: our evidence points to the latter. The decomposition shows these models possess competence (o3: 94.9% under Auto-Fulfill) but fail at sharing (15.2% under Auto-Request). Reasoning traces reveal models constructing competitive rationales in a setting where such strategies are payoff-irrelevant, suggesting **pattern-matched heuristics from training**. The micro-incentive result confirms this: a 10% sender bonus nearly triples o3's performance (+190.7%), showing these models default to requiring individual payoffs even when instructions explicitly direct otherwise.
>
> ## Environment Generalizability
>
> We validated our findings in a second zero-cost environment with a different cooperation mechanism (temporary access that expires), making delays immediately costly and cooperation opportunities self-evidently valuable. **The same qualitative inversion reproduces**: relative to the perfect baseline, Claude Sonnet 4 reaches 75.0% of optimal performance, o3-mini 39.1%, and o3 22.8% (details in our response to **Reviewer o2c5**).
>
> The benchmarks suggested in [1]-[3],[7] introduce mutual dependence, private costs, etc, confounds that make it harder to isolate whether agents cooperate when private payoffs provide no reinforcement. Our claim concerns what happens when such reasons are absent. The suggested benchmarks test complementary questions, whether cooperation improves when agents have instrumental reasons to cooperate, but **cannot isolate the phenomenon we study**. Controlled variation within the zero-cost regime is therefore the appropriate generalization axis.
>
> We note that [4] and [5] independently support this: both find that capable models do not cooperate from instructions alone when payoffs are flat, across different environments. We will expand the limitations section to discuss opportunity costs and prompting frameworks like Reflexion [6].
>
> ## Comparison between Strong/Weak models
>
> Thank you for this question: we compared reasoning traces across all baseline runs for o3 and o3-mini (50 agents each). The divergence is immediate: **37/50 o3 agents adopt withholding or market-framing modes from round 1, versus 0/50 for o3-mini**, which universally starts in cooperative or literal task-execution modes. Of the 48/50 o3 agents exhibiting withholding, **25 do so before any information exchange occurs**, showing the strategic frame is front-loaded from the initial task, not from interaction. o3-mini never transitions into withholding; its failures are stalled execution loops, not strategic choices.
>
> ## Prompt Ablations
>
> We conducted prompt ablations, removing corporate framing, replacing economic language with neutral/prosocial terms, and rephrasing the goal instruction. Our findings persist; full results are in our response to **Reviewer TDtb**.

---

> > ### Author Rebuttal · Reviewer_Z3rN · 2026-04-03
> >
> > I appreciate the authors providing the additional prompt ablations. However, the rebuttal does not resolve the core methodological and foundational concerns raised in my initial review.
> >
> > # Unresolved Issues from the Rebuttal
> >
> > **1. Unaddressed Core Question**
> >
> > My primary critique regarding prompting was whether advanced, multi-turn reasoning frameworks such as Reflexion or iterative self-correction [6] could mitigate this cooperative failure, as prior work [8] shows prompting architectures heavily impact multi-agent performance. Ablations A, B, and C only test zero-shot semantic rephrasing (e.g., swapping "dollars" for "points"). They do not test whether an agent can reflect on and self-correct its withholding strategy. By omitting this baseline, it remains **unclear if the observed behavior is a fundamental reasoning failure or simply a limitation of single-pass, zero-shot prompting**.
> >
> > **2. Incomplete Model Reporting**
> >
> > The main paper evaluates eight different models, yet the ablation table only provides data for three. Excluding the other five models (including the top-performing Gemini-2.5-Pro and DeepSeek-R1) makes it difficult to fully assess the variance that semantic framing introduces across the broader model landscape.
> >
> > **3. Economic Framing Confounders (Ablation B)**
> >
> > The authors state the ablations show their findings are not an artifact of economic framing. However, under Ablation B (removing economic terms), the o3 model's performance improves from 34.4 to 52.0, which is an approximate 51% increase. This significant jump suggests that the default economic/market framing is, in fact, a major confounder driving o3's defection logic.
> >
> > **4. Misalignment Between Terminology and Statistical Findings** The rebuttal concludes by referencing a "core capability-cooperation inversion." However, the authors concede that capability and cooperation are statistically uncorrelated ($r=0.16, p=0.71$). Framing a lack of statistical correlation as a causal "inversion" misrepresents the paper's own findings.
> >
> >
> >
> > # Fundamental Concerns
> >
> > - **Incremental Contribution:** The underlying finding that LLMs struggle to cooperate without structured incentives is already well-established. As the authors acknowledge, prior work ([1], [4], [5]) demonstrates reasoning models free-riding, defecting, and failing at zero-shot coordination. While the specific testbed (a sender-neutral environment) is creative, constructing a new environment to demonstrate a phenomenon the literature already widely accepts (that LLMs struggle to cooperate without explicit structural incentives, cooperative mechanics or scaffolding) represents an incremental finding rather than a novel scaling failure.
> > - **Overstated Conclusions:** The observed withholding appears heavily influenced by the corporate-themed prompt (as evidenced by o3's 51% improvement without economic terms). Claiming these models are fundamentally uncooperative without testing agentic self-correction loops, and introducing a new "inversion" scaling law that is unsupported by the reported p-values ($p=0.71$), draws a much stronger conclusion than the evidence shows.
> >
> >
> >
> >
> >
> > # Additional References
> >
> > [8] Dang, Y., Qian, C., Luo, X., Fan, J., Xie, Z., Shi, R., ... & Sun, M. (2025). Multi-agent collaboration via evolving orchestration. *arXiv preprint arXiv:2505.19591*.

---

> > > ### Author Response · Authors · 2026-04-07
> > >
> > > We thank the reviewer for their continued engagement.
> > >
> > > ## Terminology
> > >
> > > Our aggregate statistic (r=0.16,p=0.71) **does not support a general negative scaling law** between capability and cooperation. Our claim is: **capability does not reliably predict cooperation**, and there are pairwise cases (e.g., o3 vs. o3-mini) where the more capable model cooperates less. To avoid ambiguity, we will revise “capability-cooperation inversion” to “**capability-cooperation gap**”.
> > >
> > > ## Ablation B and Economic Framing
> > >
> > > Ablation B shows a meaningful framing effect for o3: removing economic language improves performance from 34.4 to 52.0. Our conclusion is narrower: this **does not eliminate the central cross-model gap** or the underlying strategic interpretation. Under the same neutralized framing, o3 reaches 52.0, **still only 25.5% of perfect play**, while o3-mini reaches **105.0 (51.5% of optimal)**, more than double o3’s output.
> > >
> > > Analyzing o3's reasoning under Ablation B, the core market-style frame persists. **All 10/10 agents** exhibit both market framing and explicit strict-withholding language; 5/10 reach their first such thought by round 2, and 4/10 do so before any information exchange occurs.
> > >
> > > Bargaining remains, but becomes less obstructive. Late-round non-submission thoughts remain bargaining-heavy (84.4%), but cooperative/trust language rises to 65.6% (vs. 39.1% at baseline), while waiting language falls to 59.4% (vs. 70.0%). This is consistent with the aggregate improvement: prompt wording modulates the severity of the failure, but it does not remove the underlying strategic frame, nor does it explain why a more capable model still underperforms its smaller counterpart under the same neutralized prompt.
> > >
> > > ## Reflexion and Self-Correction Frameworks
> > >
> > > Several existing results already constrain the explanation that this behavior is simply a limitation of single-pass prompting. First, our policy-level intervention goes beyond semantic rephrasing by explicitly specifying the optimal behavioral protocol (“request what you need; send when asked; submit immediately”), yet cooperation-limited models still substantially underperform. Second, under Auto-Fulfill, these same models achieve **>90% of optimal**, showing that they can execute the task when the sharing bottleneck is removed. Third, micro-incentives nearly triple o3’s performance without changing the prompting architecture, indicating that cooperation shifts sharply when individual payoffs change under otherwise identical prompting.
> > >
> > > Taken together, these findings make a **purely single-pass prompt-phrasing account unlikely**. Our reasoning analysis further suggests that withholding is often treated as strategically coherent rather than as an obvious mistake, which helps explain why simple reflection may not fully resolve the behavior.
> > >
> > > Reflexion may improve performance, but that would answer a related but different question: how much behavior can be recovered through added orchestration, rather than what the baseline model treats as instrumentally rational in this sender-neutral setting. We will clarify this distinction in revision.
> > >
> > > ## Incomplete Model Reporting
> > >
> > > We ran three models under rebuttal-time compute constraints, selected to span the main failure modes: a strong cooperator (Claude Sonnet 4), a cooperation-limited model (o3), and an intermediate case (o3-mini). Full ablation results for all eight models will be included in the revision.
> > >
> > > ## Incremental Contribution
> > >
> > > We respectfully disagree with this characterization. Prior work demonstrates cooperation failures in settings with private costs [5], reputation dynamics [4], or coordination complexity [1,2,3,7]: settings in which non-cooperation can be strategically rationalized. Our contribution is to show that a cooperation bottleneck persists even in a **sender-neutral regime** designed to largely remove direct payoff rationalizations for withholding: helping is costless to the sender, instructions are explicit, and withholding is weakly dominated under every plausible objective.
> > >
> > > In addition, the controlled decomposition framework, isolating cooperation from competence via selective automation, is, to our knowledge, new. The finding that models achieving **>90% competence under Auto-Fulfill** can simultaneously achieve **<20% cooperation under Auto-Request** provides a dissection that existing benchmarks do not isolate, and the targeted interventions that follow from it are directly actionable for multi-agent system design.
> > >
> > > ## Conclusion
> > >
> > > To avoid overstating scope, we will revise the framing. We **do not claim a general negative scaling law** or that richer scaffolds cannot improve outcomes. Our claim is that in sender-neutral settings, capability does not reliably predict cooperation; the **controlled decomposition** isolates a cooperation bottleneck beyond competence; and targeted interventions show it is sensitive to incentive structure and failure-mode-specific fixes.

---

### Official Review · Reviewer_TDtb · 2026-03-13

**Soundness:** 3
**Presentation:** 4
**Significance:** 3
**Originality:** 4
**Overall Recommendation:** 5
**Confidence:** 3

**Summary:**

This paper investigates the cooperative behavior of LLM agents in zero-cost multi-agent collaboration scenarios. The authors construct a turn-based information exchange environment to test whether agents will cooperate when helping others incurs no private cost but generates substantial collective benefits. Experiments reveal that highly capable models (e.g., OpenAI o3) sometimes significantly underperform smaller models (e.g., o3-mini) and exhibit non-cooperative behaviors. By implementing a causal decomposition experiment, the paper successfully disentangles agents' competence failures from cooperation failures. Furthermore, the authors test interventions such as policy-level instructions, micro-incentives, and limited visibility, demonstrating that low-cost methods can effectively restore the cooperative willingness of large models.

**Compliance With Llm Reviewing Policy:**

Affirmed.

**Final Justification:**

The Rebuttal has addressed my concerns, and I will accordingly increase my score to 5.

**Key Questions For Authors:**

**Questions**

- How sensitive are the agents to the initial system prompts? Could a slight modification in the phrasing of the overall objective instruction eliminate the unnecessary zero-sum game mentality?

- In the Agent Reasoning Analysis, the authors utilized regular expressions. Have the authors considered using another powerful LLM as a judge to evaluate the semantic intent of the private thoughts, replacing the rigid keyword matching? I am curious whether the cooperation failures in models like o3 are solely related to selfishness or stem from other inherent flaws in the model itself. Using an LLM to analyze the reasons for failure might be a good idea.

- A very peculiar phenomenon is that even with the addition of various interventions, o3's Task Completed rate remains quite low. What is the reason for o3's continued failure after interventions are applied? Is it still due to a disinclination to cooperate, or are there other reasons beyond cooperation?

- Does the prompt for private thoughts induce competitive language? If the private_thoughts field is completely removed, would the behavioral patterns change?

**Limitations:**

yes

**Strengths And Weaknesses:**

**Strengths**

- **Ingenious decoupled experimental design:** Through the automated mechanisms of Auto-Request and Auto-Fulfill, the paper isolates the agents' capabilities and intentions. It effectively investigates the root causes of cooperation failures.

- **Findings with significant scientific and social impact:** The study reveals that scaling laws may fail in the dimension of collaboration (e.g., o3's task completion rate is only 17% of perfect play). Given the current industry pursuit of stronger models and extensive multi-agent collaboration, this research serves as an important caution. I appreciate such discoveries in this paper.

- **Practical interventions:** The paper goes beyond identifying problems by providing actionable solutions. For instance, offering a tiny sharing bonus equivalent to 10% of the task value can drastically improve the performance of models exhibiting cooperation bottlenecks.

---
**Weaknesses**

- **Limited complexity and generalizability of the test environment:** The test environment is a highly structured, turn-based game. This may not fully capture the complexity of unstructured natural language collaboration in the real world (such as code reviews or asking for help in daily conversations). In such real-world tasks, agents might naturally recognize the necessity and benefits of cooperation.

- **Relatively crude internal reasoning analysis:** The Agent Reasoning Analysis relies heavily on regular expression-based keyword matching (e.g., matching "leverage", "bargain", etc.). This hard-coded pattern matching could miss more subtle, deceptive behaviors by the models or misclassify benign thought processes.

---

> ### Author Rebuttal · Authors · 2026-03-31
>
> We thank the reviewer for their positive assessment and thoughtful questions.
>
> ## Prompt Ablations
>
> The reviewer's question about whether slight prompt modifications could eliminate competitive framing motivated us to run controlled ablations by re-configuring the baseline prompt in three ways that address unique confounds:
>
> **A**: Remove potential corporate framing (company name, rename prompt items)
>
> **B**: Everything in A + rename economic terms (dollar to points, tasks become generic, etc)
>
> **C**: Rephrase agent instruction that defines the goal
>
> | Model | Baseline | A | B | C |
> | :--- | :--- | :--- | :--- | :--- |
> | o3-mini | 102.8 | 111.0 | 105.0 | 82.0 |
> | o3 | 34.4 | 38.0 | 52.0 | 24.0 |
> | Claude Sonnet 4 | 132.0 | 128.0 | 133.0 | 109.0 |
>
> **Ablations A and B preserve the main findings**, suggesting that the results are not an artifact of corporate/economic framing alone. Ablation C tests the reviewer's hypothesis: rephrasing the objective instruction does not eliminate the zero-sum mentality. These results suggest that while prompt phrasing influences overall performance, it does not explain away the **core capability–cooperation inversion**.
>
> ## LLM as a Judge for Reasoning Analysis
>
> This suggestion proved valuable. We ran a semantic audit using GPT-5.4 (temperature=0, JSON rubric) as an independent judge over a stratified sample of 569 private-thought examples across baseline transcripts. Each thought was judged in context (recent agent state) and assigned to one of four labels: withholding, cooperative, competence_issue, or ambiguous.
>
> The paper's reasoning claims are supported. Hard-defection cases (Table 7) persist with the LLM-judge: **73.2% remain withholding** and **false negatives were minimal: only 6.2%** of regex-negative samples were reclassified as withholding. The audit reveals model-level distinctions previously invisible to regex: o3 has 33.3% of conditional thoughts judged as withholding, versus 3.8% for GPT-5-mini and 0% for GPT-4.1-mini and Gemini-2.5-Pro.
>
> We emphasize that the paper's central claims rest on behavioral decomposition (Table 2, Figure 4), which is independent of reasoning-trace interpretation; these findings serve as confirmatory evidence and will be included in the revised manuscript.
>
> ## o3 after Incentives
>
> Thank you for this question. o3 actually shows the largest response to micro-incentives of any model (+190.7%), but the reviewer's observation that a residual gap to perfect play remains is correct and reveals a layered failure structure: **cooperation failure was masking coordination inefficiency underneath**.
>
> With incentives, cooperative language in o3's late-round thoughts rises sharply (65.9% vs. 20.7% at baseline) and agents explicitly reference the sharing bonus; yet thoughts remain saturated with bargaining (\~91%) and contingent planning (\~64%), deferring execution through negotiations. This reveals layered failures: cooperation failure masks an underlying coordination inefficiency. Incentives resolve the first bottleneck and expose the second: o3 begins sharing but still wraps exchanges in negotiation overhead that delays execution. This suggests a natural next step: execution-targeted interventions.
>
> ## If private_thoughts are removed, does behavior change?
>
> This is a sharp observation. Our limited-visibility intervention already removes agents' private thoughts alongside other signals, so inference from it is indirect but informative. If the private_thoughts field were inducing competitive language that drove non-cooperative behavior, removing it should substantially restore cooperation; yet **o3 improves by only 22.1% under limited visibility compared to 190.7% under micro-incentives**, and its cooperation rate in the Auto-Request condition (15.2%) remains nearly unchanged. We interpret private thoughts as a diagnostic window into decisions driven by other factors rather than a cause of those decisions. We will clarify this relationship in the revised manuscript.
>
> ## Limited generalizability of environment
>
> The reviewer hypothesizes that in more naturalistic settings (code reviews, asking for help) agents might naturally recognize the benefits of cooperation and behave accordingly. To test this, we validated our findings in a new environment modeling authorization workflows, where agents coordinate on time-sensitive deliverables by granting temporary access to resources. Cooperation opportunities expire, so delays cause permanent loss, creating the kind of immediate, recognizable benefit the reviewer suggests should promote cooperation.
>
> Despite this urgency and a less rigid interaction structure than the original environment, **the same qualitative pattern persists**: Claude 4 Sonnet reaches 75.0% of optimal performance, o3-mini 39.1%, and o3 22.8%. Full results are included in our response to **Reviewer o2c5**. This suggests the phenomenon reflects model-level behavioral tendencies rather than the original environment’s specific structure.

---

> > ### Author Rebuttal · Reviewer_TDtb · 2026-04-01
> >
> > Thank you for the response! Your clarifications have addressed my concerns, and I will increase my score to 5 accordingly. I look forward to seeing the revised version and wish you the best of luck with your work!

---

> > > ### Author Response · Authors · 2026-04-02
> > >
> > > Thank you for the thoughtful follow-up and for updating your assessment! We’re very glad the rebuttal addressed your concerns, and we appreciate your careful engagement with the paper.

---

### Decision · Program_Chairs · 2026-04-30

**Decision:**

Accept (regular)

**Comment:**

Three reviewers think the paper should be accepted, one that it should be rejected. The central concerns raised by the reviewer who proposed rejection were four. The first and the fourth were both about the connection between this work and existing literature. The second asked for a comparison of reasoning traces between high reasoning models and smaller models, and the third asked for further investigation into prompting strategies.

The authors provide a detailed response as far as the relationship between their work and prior work goes, that I find satisfactory. They also provide comparisons between strong and weak models, as well as prompt ablations. In these respects, I think that the authors don't really respond to the reviewer's concern very directly. On the other hand, other reviewers raised similar concerns and were satisfied by the author's response.

The critical reviewer instead shifts to a more "fundamental" concern that the contribution is incremental and the conclusions are overstated. The authors accept that the conclusions were in some places overstated. I think that their contribution is indeed somewhat incremental. I suspect that the results of this experiment will not prove especially predictive of the cooperation behavior of more capable models, and that as a result, this gives us only a snapshot of a moment in time as opposed to a real guide for future development of cooperative AI systems. So I am sympathetic to the perspective of the critical reviewer. However, not so much so that I would overrule the three reviewers who thought this a clear accept.